# Male childlessness as independent predictor of risk of cardiovascular and all-cause mortality: A population-based cohort study with more than 30 years follow-up

Angel Elenkov[1,2]*, Aleksander Giwercman[1,2], Sandra Søgaard Tøttenborg[3], Jens Peter Ellekilde Bonde[3,4], Clara Helene Glazer[5], Katia Keglberg Haervig[3], Ane Berger Bungum[3], Peter M. Nilsson[6]

1 Department of Translational Medicine, Molecular Reproductive Medicine, Lund University, Malmoe, Sweden, 2 Reproductive Medicine Center, Skåne University Hospital, Malmoe, Sweden, 3 Department of Occupational and Environmental Medicine, Bispebjerg Hospital, Copenhagen, Denmark, 4 Department of Public Health, The Faculty of Health Sciences, University of Copenhagen, Copenhagen, Denmark, 5 Department of Urology, Rigshospitalet, Copenhagen, Denmark, 6 Department of Clinical Sciences, Internal Medicine Research Group, Skåne University Hospital, Malmoe, Sweden

* angel.elenkov@med.lu.se

**Data Availability Statement:** The data underlying the results presented in the study can be regarded

## Abstract

In a recent population-based study, an elevated risk of the Metabolic syndrome (MetS) and type 2 diabetes was found in childless men compared to those who have fathered one or more children. Therefore, by using a larger cohort of more than 22 000 men from the Malmo Preventive Project (MPP) we aimed to expand our observations in order to evaluate the metabolic profile of childless men and to evaluate if childlessness is an additional and independent predictor of major adverse cardiovascular events (MACE), mortality and incident diabetes when accounting for well-known biochemical, anthropometric, socio-economic and lifestyle related known risk factors. Logistic regression was used to assess risk of MACE, diabetes and MetS at baseline. Multivariate Cox regression was used to evaluate the risks of MACE and mortality following the men from baseline screening until first episode of MACE, death from other causes, emigration, or end of follow-up (31st December 2016) adjusting for age, family history, marital status, smoking, alcohol consumption, educational status, body mass index, prevalent diabetes, high blood lipids, increased fasting glucose and hypertension. Childless men presented with a worse metabolic profile than fathers at the baseline examination, with elevated risk of high triglycerides, odds ratio (OR) 1.24 (95% CI: 1.10–1.42), high fasting glucose OR 1.23 (95%CI: 1.05–1.43) and high blood pressure, OR 1.28 (95%CI: 1.14–1.45), respectively. In the fully adjusted prospective analysis, childless men presented with elevated risk of cardiovascular mortality, HR: 1.33 (95% CI: 1.18–1.49) and all-cause mortality, HR 1.23 (95%CI: 1.14–1.33), respectively. In conclusion, these results add to previous studies showing associations between male reproductive health, morbidity and mortality. Male childlessness, independently of well-known socio-economic, behavioral and metabolic risk factors, predicts risk of cardiovascular disease and

as potentially sensitive since the majority is derived from national registries. Dataset access is controlled by scientific committee. Request for access and applications can be sent to the principal investigators as described here: https://www.med. lu.se/malmoe_kost_cancer_och_malmoe_ foerebyggande_medicin/malmoe_foerebyggande_ medicin.

**Funding:** The study was supported by grants: Interreg V funded program ReproUnion - all authors, joint European Association of Urology – ReproUnion scholarship - Angel Elenkov Research Council of Sweden Peter M Nillson The funders had no role in study design, data collection and analysis, decision to publish, or preparation of the manuscript.

**Competing interests:** The authors have declared that no competing interests exist.

mortality. Consequently, this group of men should be considered as target population for preventive measures.

## Introduction

In recent years, studies from the United States and Europe have shown associations between impaired male reproductive health and risk of various non-communicable adult-onset diseases. Low sperm counts have been associated with higher risk of type 2 diabetes, metabolic syndrome (MetS) and all-cause mortality [1–4]. It has also been shown that men who fathered children following intracytoplasmic sperm injection (ICSI)–which is the preferred method when semen quality is poor—more often is prescribed drug treatment for hypertension, dyslipidemia, and MetS [5]. Married men who have not fathered children, which might reflect impaired fertility, have presented with higher risk of diabetes and cardiovascular disease [6, 7]. Therefore, a man's reproductive performance was suggested to be a general health marker [4].

The biological link between impaired male reproductive function and general health is complex and probably includes an interplay between genetic and environmental factors possibly operating as early as during intrauterine life [8–11]. As a consequence, testicular insufficiency and testosterone deficiency may serve as a possible mediator between these associations [12–15]. Since family size is often influenced by fecundity, men with lower offspring number (i.e. one child or childless) may represent a group with overrepresentation of testicular dysfunction [6, 16, 17].

In a recent study of 2572 Swedish men, an elevated risk of MetS and type 2 diabetes was found in childless men as compared to those men who have fathered one or more children [7]. Having access to a larger cohort of men from background population and a longer follow-up, we wanted to evaluate the metabolic profile of childless men and estimate the risks of major adverse cardiovascular events (MACE) and cardiovascular and all-cause mortality accounting for well known risk factors of cardiovascular disease (CVD) including smoking, alcohol consumption, high body mass index (BMI), dyslipidemia, hypertension, family history of CVD high fasting glucose, and socioeconomic factors. The aim was to investigate whether male childlessness *per se* predicted an additional risk increase. Since previous research has pointed to different causation as considering stroke events [18] a separate end point was evaluated excluding the latter from the composite MACE end point (referred to as non-stroke MACE).

## Subjects and methods

### Study population

Data for this study was obtained from the Malmo Preventive Project (MPP). The original project was designed to screen for cardiovascular risk factors and alcohol overuse in the background population living in Malmo, southern Sweden. Baseline examination was conducted between 1974 and 1984 at the Department of Preventive Medicine, Malmo University Hospital [19]. A total of 22 444 men aged 25 to 63 years were recruited with a participation rate of approximately 71%.

The physical examination and laboratory investigations have previously been described in detail [19–21]. In brief, all subjects were examined for weight (kg) and height (m), and body mass index (BMI; kg/m$^2$) was calculated. Blood pressure was measured in mmHg after 10 minutes' rest. Blood samples were drawn after an overnight fast and analyzed for fasting blood glucose, total serum cholesterol, and serum triglycerides using standard methodology at the

Department of Clinical Chemistry. A self-administered questionnaire with 260 questions focusing on family and medical history, as well as lifestyle (smoking habits, physical activity), and educational level was completed.

## Information on paternal status

Information regarding paternal status was gathered using two sources: (a) the baseline questionnaire where men answered "yes" or "no" to the question *"Do you have children?"* and (b) the Swedish Tax Agency Statistics (STAS) which presents the number of children and their personal ID (date of birth) to each father as a consecutive number linking data using unique personal identification code assigned to all Swedish citizens. The data was last updated at the end of follow up (31st December 2016).

The study has been approved by the Regional Ethics Review Board at Lund University (85/2004) and complies with the declaration of Helsinki. Participants gave informed consent.

## Retrieval of cardiovascular disease endpoints and mortality

Information on deaths and cardiovascular events during follow-up were obtained through cross-linking the cohort members with the following population registers: The Swedish National Hospital Discharge Register, the Swedish National Coronary Angiography and Angioplasty Registry (SCAAR), the Swedish National Cause of Death Register (SNCDR), the local Stroke Register in Malmo, and the National Swedish Classification Systems of Surgical Procedures. The registers have recently been validated for classification of outcomes and found to be of high quality with validity 86–97% [22]. Information on mortality including age and date of death were retrieved from SNCDR.

Cardiovascular mortality was defined as a separate end point deriving data from SNCDR.

MACE was defined as fatal or non-fatal myocardial infarction, fatal or non-fatal stroke, death due to ischemic heart disease, coronary artery bypass graft surgery (CABG), or percutaneous coronary intervention (PCI). Transient ischemic attack was not included as a stroke event.

Non-stroke MACE was defined as a separate endpoint in order to determine whether childlessness modifies the risk of other types of MACE except stroke events [18].

Data for the aforementioned endpoints was sourced as follows: Myocardial infarction was defined on the basis of ICD-9 code 410 and ICD-10 code I21; Death due to cardiovascular disease (CVD) was defined on the basis of codes 412 or 414 (ICD-9) and I22–I23 or I25 (ICD-10); CABG was identified from national Swedish classification systems of surgical procedures and defined as procedure codes 3065, 3066, 3068, 3080, 3092, 3105, 3127, or 3158 (the Op6 system) or procedure code FN (the KKÅ97 system); PCI was identified from SCAAR. Stroke was defined on the basis of codes 430–432, or 434 (ICD-9) and code I60-I63 (ICD-10).

## Identification of diabetes

In total, 15 different sources of data were used to identify individuals with diabetes. All individuals in the following registers were searched: Diabetes 2000 (except certain with missing data), the All New Diabetes cases in Scania (ANDIS), and the Swedish National Diabetes Register (SNDR) were considered to have diabetes, as well as men with at least two $HbA_{1c}$-values $\geq 6.0\%$ (not on the same day) in the $HbA_{1c}$ register at Department of Clinical Chemistry, Malmö. In the National Patient Register (inpatient care and outpatient care) and the Swedish Cause of Death Register from the National Board of Health and Welfare, individuals with the ICD10 codes E10-E14 and O244-O249 (and corresponding ICD7-9 codes), were treated as men with diabetes. The ATC code A10 was used to identify individuals with diabetes in the

Prescribed Drug Register. Further, men who in the baseline examination had fasting blood glucose ≥ 5.6, or post-load glucose ≥11 mmol/L at 120 minutes oral glucose tolerance test, or had reported intake of ACT category A10 drugs, or have responded "*Yes*" on the question "*Do you have diabetes*?" was considered to have diabetes.

**Co-variates and data retrieval.** Data on marital status (never-married; married; divorced; widowed), number of cigarettes per day (none; 1–10;10–20;30–40; 40 or more), and educational level (no education; primary school; secondary school or more) was derived from the baseline questionnaire. A positive family history for CVD was defined as reporting having at least one first-degree relative with MI before the age of 60 years as described previously [23]. No family history was defined as no reported first-degree relatives with MI before 60 or no answer in the questionnaire. The co-variate "alcohol consumption" was determined based on questions on alcohol habits according to "Malmo modification of the brief Michigan Alcohol Screening Test" (Mm-MAST). It is described elsewhere and contains seven questions [24]. The questions analyzed aim to give a quantitative measure of the risk habits associated with alcohol consumption. They were divided into two categories containing confirmative answers to three questions each. "Moderate risk alcohol habits" includes the following three questions: *"Do you usually have a drink before going to a party?", "Do you mostly drink alcohol, e.g. a bottle of wine, at weekends or holidays?",* and *"Do you drink a couple of beers, some glasses of wine or a drink to relax on a daily basis?".* The category "heavy risk alcohol habits" includes the following three questions: *"Do you tolerate liquor better now than 10 years ago?", "Has it ever happened that after a party you do not remember how you got into bed?",* and *"Do you usually have bad consciousness after a party*?" If the answer was "yes" to one or more of the questions included in either of the two drinking categories the total answer to that category was considered as "yes". If the answer was "no" to all of the abovementioned question the subject was classified to third category—"low risk alcohol consumption". Three separate MetS components were selected as continuous co-variates: fasting blood glucose, triglycerides, blood pressure after 10 minutes rest, available from the baseline screening.

## Statistical methods

**Baseline morbidity.** We categorized a man as childless if he answered "no" in the questionnaire and had no registered children in STAS. Men who answered "yes" in the questionnaire and/or had children registered in STAS were categorized as fathers. Men with missing data in both the questionnaire and STAS were excluded, as were men who answered "no" in the questionnaire but had STAS registered children before the entry date (conflicting information).

Baseline characteristics were compared by T-test for continuous variables and Chi-square test for categorical. The associations between childlessness and high fasting glucose, high blood pressure, and high triglycerides were assessed using logistic regression analyses. NCEP III criteria were used to determine the threshold values [25]. To account for possible confounding, the models were adjusted for age, marital status, educational level, BMI, alcohol consumption and smoking.

Prevalent diabetes and MACE risks were evaluated at baseline using the same methodology as described above.

**Prospective analysis.** The risk of having an incident MACE was evaluated using multivariate Cox regression and presented with hazard ratios (HR) and corresponding 95% confidence intervals (95%CI) using time on study (in years) as the underlying time scale following men from baseline screening until first episode of MACE, death from other causes, emigration, or end of follow-up on 31st December 2016 adjusting for age at baseline as a continuous co-

variate. Analysis was accompanied with Kaplan-Meyer curves. The proportional hazards assumption was upheld for all analyses as assessed using log minus log plots. In order to explore associations with the metabolic disturbances separately, analyses were adjusted in two steps: *Model 1*: age, marital status, educational level, alcohol consumption and smoking, and *Model 2*: Model 1 + BMI, prevalent diabetes, fasting blood glucose, triglycerides, blood pressure, family history of CVD.

To avoid introducing immortal time bias in the survival analysis the men were grouped according to fatherhood status based on baseline status (start of follow up). Childlessness at the start of follow up (baseline) is related to age. In order to improve the use of childlessness as a proxy of infertility, only men 45 years or older were included in the analysis (N = 11343; 51.5% of all) classifying them according to fatherhood status at baseline.

Cases with prevalent MI (n = 97) or stroke (n = 25) were excluded from prospective analysis evaluating risk estimates for MACE and CVD mortality. The same model was used to study the risk of all-cause mortality, defining mortality as death before the end of follow-up.

We also used Cox regression analysis to calculate HRs for incident diabetes with the same adjustment models as described above. A total of 683 men had prevalent diabetes and were excluded from prospective analysis when evaluating the risk of developing diabetes.

As childlessness could be related to not having a partner rather than impaired fertility, a sensitivity analysis was performed including only married men [7].

All analyses were performed using SPSS v.25.0 (IBM corp., 2017, SPSS Statistics for Macintosh) using confidence intervals to interpret the results.

## Results

### Background characteristics

**Entire cohort.**  A total of 2943 men remained childless and 17 729 were fathers. In total, 1773 men were excluded due to missing data as were 357 who had conflicting information on the number of children at time of baseline. The mean (SD) follow-up time for the cohort was 30.7 (9.4) years.

**Men 45 years and older.**  A total of 2134 men 45 year or older at baseline were childless and 9209 were fathers. Of men categorized as childless at baseline fifteen fathered a child after the start of follow up. Mean (SD) age of the childless men was 48.9 (4.1) years compared to 48.7 (3.9) years among the fathers. Baseline characteristics are presented in Table 1.

**Baseline morbidity.**  In the adjusted model, childless men presented with a worse metabolic profile than fathers at the baseline examination, with higher risk of high triglycerides, elevated fasting blood glucose and high blood pressure (Table 2). In the sub-analysis including only married men results remained unchanged (Table 2).

No increased risk of prevalent diabetes was observed among childless men than in fathers, (OR 1.15; 95% CI: 0.89–1.49). The childless men and fathers did not differ as considers prevalence MACE at baseline (OR 1.31; 95%CI: 0.68–2.50).

**Prospective analysis.**  In the prospective analysis, men childless at baseline, in the fully adjusted model, presented with higher risk of cardiovascular mortality, HR: 1.33 (95% CI: 1.18–1.49) and all-cause mortality, HR 1.23 (95%CI: 1.14–1.33, Fig 1). After adjustment the childless men showed higher risk of incident cases of MACE, despite not significantly different from 1 (Table 3). The majority of risk estimates remained unchanged in the sub-analysis including only married childless men (Table 3).

For diabetes, in the prospective model, no increased risk was detected: HR 0.97 (95%CI: 0.83–1.13).

**Table 1. Socio-demographic and lifestyle characteristics of men older than 45 years with and without children at baseline.**

| | Childless men (n = 2134) | Fathers (n = 9209) | p-value |
|---|---|---|---|
| **Marital status** | | | 0.001 |
| • **Never married (%)** | 43.8 | 1.3 | |
| • **Married, Divorced, Widower (%)** | 56.2 | 98.7 | |
| **Level of education** | | | 0.01 |
| • **No education (%)** | 11 | 4.1 | |
| • **Primary school (%)** | 62.6 | 63.6 | |
| • **Secondary school (%)** | 26.3 | 32.3 | |
| **BMI (kg/m$^2$): Mean (SD)** | 25.3 (3.8) | 25 (3.2) | 0.001 |
| **Alcohol consumption** | | | 0.03 |
| • Low risk alcohol consumption | 51.9 | 48.9 | |
| • Moderate risk alcohol habits | 26.7 | 29.9 | |
| • High risk alcohol habits | 21.2 | 21.1 | |
| **Present smoker** | | | 0.08 |
| • **Non-smoker (%)** | 62.3 | 59.4 | |
| • **10–20 cig. /day (%)** | 21.2 | 24.4 | |
| • **20–30 cig. /day (%)** | 12.5 | 12.2 | |
| • **30–40 cig./day (%)** | 2.7 | 2.9 | |
| • **More than 40 cig./day (%)** | 1.1 | 1.1 | |
| **Age** | 48.9 (4.1) | 48.7 (3.9) | 0.08 |
| **Cholesterol (mmol/L)** | 5.81 (1.35) | 5.76 (1.04) | 0.06 |
| **Tryglicerides (mmol/L)** | 1.59 (0.95) | 1.57 (1.01) | 0.47 |
| **Serum glucose (mmol/L)** | 5.01 (1.19) | 4.93 (1.07) | 0.007 |
| **Family history of CVD (%)** | 0.61 | 0.58 | 0.68 |

Results presented with means (SD) and proportions (%). The results of the univariate analysis are presented with p-values.

## Discussion

In the present study, we found that childless men, at baseline health screening being on average slightly above 45 years, presented with statistically significantly higher risk of metabolic risk profile associated with MetS. Being childless was also associated with a higher risk of cardio-vascular, and all-cause mortality when compared to men who were fathers. Results were robust

**Table 2. Odds Ratio (OR) with 95% confidence intervals (95% CI) for metabolic syndrome components in childless men compared to fathers (reference group) at baseline adjusted for age, marital status, educational level, BMI, alcohol consumption, smoking.**

| | | *Childlless men n/total (%)* | *Fathers n/total (%)* | **OR[95% CI]** |
|---|---|---|---|---|
| **Including all childless men** | **Hyperglycaemia** * | 306 / 2114 (14.5) | 1074/9176 (11.7) | 1.27 [1.09–1.49] |
| | **Hyperlipidemia ‡** | 687/ 2126 (32.3) | 2816/9198 (30.6) | 1.11 [0.98;1.25] |
| | **Elevated BP §** | 885/2114 (41.9) | 3235/9201 (35.7) | 1.26 [1.12;1.42] |
| **Including only married childless men**** | **Hyperglycaemia** * | 172/ 1118 (15.4) | 1052/ 9057 (11.6) | 1.29 [1.05;1.58] |
| | **Hyperlipidemia ‡** | 383/1192(32.1) | 2780/9079 (30.6) | 1.16 [0.99;1.36] |
| | **Elevated BP §** | 492/ 1197 (41.3) | 3193/9082 (35.2) | 1.28 [1.10;1.49] |

* Hyperglycemia defined as a fasting blood glucose level $\geq$ 5.6 mmol/L

‡ Hyperlipidemia defined as Triglycerides$\geq$1.7 mmol/L

§ Elevated blood pressure (BP) defined by BP $\geq$130/85 mmHg

**The same adjustment model is used as described above excluding marriage status.

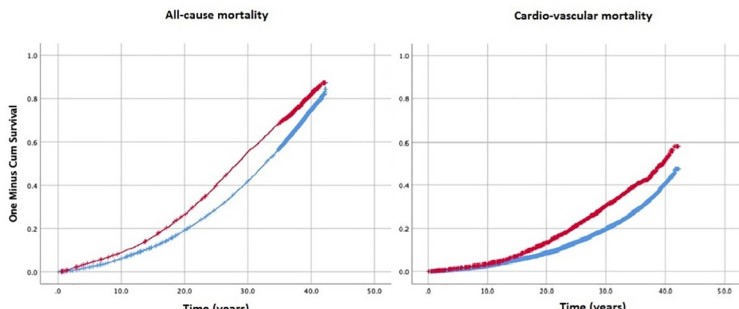

**Fig 1. Kaplan Meier survival curves of cardiovascular and all-cause mortality in childless men aged 45 years or older at baseline.as compared to fathers.** Red line denotes childless men, bue line–fathers. Log rank p value < 0.001.

to exclusion of unmarried men and thereby accounting for lack of reproductive opportunities. Interestingly, despite attenuation of HRs, the increase in mortality risks remained unchanged after adjustment for well known risk factors of CVD: BMI, prevalent diabetes, high fasting blood glucose, high triglycerides, high blood pressure, alcohol consumption, smoking, educational level and marriage status.

Thus, using childlessness as proxy for male infertility resulted in similar increased risk estimates in relation to metabolic and cardiovascular disease (CVD) as well all-cause mortality as previously reported for impaired semen quality, an established biological determinant of impaired male reproductive function [4].

The observed increased risk of MACE, CVD, and all–cause mortality among childless men is consistent with the majority of previous studies. What our study adds is the fact that mortality risks remain higher even after adjustment not only for available socio-economi, familial and behavioral factors as well as prevalent diabetes but also when adding to the model various metabolic parameters including glucose, triglycerides and hypertension. According to Eisenberg *et al* (2011) married childless men run a higher risk of dying from CVD after the age of 50 years [6]. A British study has evaluated the association between offspring number and prevalent CVD in both men and women [26]. Similar to our findings, they observed a higher risk of CVD among childless men compared to fathers. This study included 30% single men in the childless cohort which due to limited reproductive opportunities in those men might imply

**Table 3. Hazard Ratios (HR) with 95%CI for Major Adverse Cardiovascular Events (MACE) and all-cause mortality in men childless at baseline relative to fathers at baseline.** Only men 45 years or older at baseline were included.

|  | End points | Childless N/% | Fathers N/% | Crude HR | Model 1* | Model 2** |
|---|---|---|---|---|---|---|
| **Including all childless men** | MACE | 930/44% | 4047/44% | 1.15 (1.06–1.23) | 1.08(0.96–1.21) | 1.04 (0.93–1.17) |
|  | Non-stroke MACE | 715/34% | 3022/33% | 1.13 (1.04–1.25) | 1.10 (0.96–1.25) | 1.06 (0.93–1.21) |
|  | All-cause mortality | 1651/77% | 6423/69% | 1.36 (1.28–1.44) | 1.31 (1.21–1.44) | 1.28 (1.18–1.40) |
|  | Cardiovascular mortality | 724/34% | 2498/27% | 1.50 (1.37–1.65) | 1.38 (1.20–1.58) | 1.32 (1.15–1.52) |
| **Including only married childless men \*\*\*** | MACE | 52143% | 4005/44% | 1.1 (0.97–1.26) | 1.07 (0.96–1.21) | 1.04 (0.93–1.17) |
|  | Non-stroke MACE | 403/34% | 2991/33% | 1.11 (0.97–1.26) | 1.09 (0.96–1.25) | 1.05 (0.92–1.20) |
|  | All-cause mortality | 911/76% | 6334/69% | 1.34 (1.23–1.43) | 1.31 (1.20–1.43) | 1.28 (1.17–1.39) |
|  | Cardiovascular mortality | 377/31% | 2467/27% | 1.41 (1.23–1.62) | 1.37 (1.20–1.58) | 1.32 (1.15–1.52) |

* *adjusted for*: age, marital status, educational level, alcohol consumption, smoking (Model 1)

** *adjusted for*: Model 1 + BMI, *family history of CVD prevalent diabetes, fasting blood glucose, triglycerides, blood pressure (continuous variables)*.

*** The same adjustment model is used as described above excluding marriage status.

higher impact of adverse social factors rather than biological mechanisms related to the impaired reproductive function. A Swedish register-based study has previously reported that childless men have higher risk of dying from ischemic heart disease [27].

Despite previous reports showing higher prevalence of diabetes among childless men [7] our study failed to establish statistically significant association both in cross-sectional and prospective analysis.

Our findings seem to indicate that childlessnees, on top of well known risk factors of increased mortality due to CVD, is an independent predictor of this adverse life event. One can speculate whether this association is related to psychosocial mechanisms and/or a joint underlying biological cause of not fathering children and CVD. Available studies have failed to show a significant reduction in the risk of CVD mortality and all-cause mortality among childless after adjusting for chronic conditions including intellectual disabilities, schizophrenia and early disability retirement [28]. Another study based on data form The Danish IVF register has suggested that the rate of psychiatric illness is of the same order of magnitude in biological parents as in childless individuals [29]. Another important aspect is lack of reproductive opportunities. In fact, 43% of childless men in our cohort were unmarried compared to 1.3% among fathers. Furthermore, men with lower educational level were shown to be overrepresented in the childless men group in previous studies from Sweden [38] suggesting a role of psycho-social status on parity. However, in a register-based analysis from Norway CVD mortality risk among childless men remained robust to adjustment for education, region of residence, population size municipality and marriage status [30]. By adjusting for educational level, and restricting the sensitivity analysis to only married men, we have tried to account for some of these non-biological mechanisms in our study.

Previous study [2] has demonstrated that increased mortality risk is associated with impaired semen quality, regardless of whether men investigated for infertility succeed to achieve parenthood or not. Thus, biological factors related to infertility seem also to play a role for the risk of adverse health events in those men.

Since family size can be directly linked to male fertility potential [16], an overrepresentation of sub-fertility among childless men is likely to be expected. The aforementioned associations between fertility and general health are complex and various biological mechanisms have been proposed, including shared genetic traits between metabolic and reproductive pathways or *in utero* exposure to adverse environmental factors in combination with maternal health issues during pregnancy affecting both fertility and cardiovascular health [1–4, 10, 31]. Both male infertility and low testosterone (T) levels can be regarded as consequences of testicular dysfunction, and positive association between these conditions have been identified in clinical cohorts [12]. Low T levels have been associated with metabolic disturbances, diabetes, cardiovascular disease and mortality risk [32, 33]. Therefore, low T levels and impaired androgenic state might be overrepresented among childless men and can serve as a plausible mediator of the established associations in our study. Atherosclerosis, a complex and not fully elucidated condition, is among the main risk traits for the majority of cardiovascular events. Low free T levels have been associated with carotid atherosclerosis, higher carotid intima media thickness and higher coronary artery calcium score [34, 35]. Apolipoprotein A1, a biomarker for subclinical atherosclerosis, was also found to be inversely related to T levels [36].

Our study has several strengths but also limitations. Comprehensive information from Swedish national registries allows for precise information on date and cause of death, emigration, and disease diagnosis. A mean follow-up of more than 30 years makes our cohort among the ones with the longest follow-up in comparison with other similar cohort studies. Our study is representative for the background population in Sweden's third largest city allowing for inclusion and representation of men from all socioeconomic backgrounds. The meticulous

data set at baseline provides an opportunity to adjust for diabetes, BMI, blood lipids, serum glucose, blood pressure, alcohol consumption and smoking.

One of the limitations is that we could neither account for the participants´ paternal intention, nor the partner's fertility status and therefore the number of children cannot be regarded as an optimal proxy for reproductive potential. Fathers of adopted children, or those who have sired offspring with the help of assisted reproduction techniques, might have been included in the control group leading to misclassification bias. However, adoption rates in Sweden for the study period were relatively low and assisted reproduction technologies were not widely available before 1992 [38–41] so we do not believe this to be a major concern/source of bias. Only one in four childless men in Sweden has been reported to be voluntary childless [38]. Therefore the risk of our sample to reflect voluntary childlessness is limited. Furthermore, it is unknown for how long the children lived with their fathers and in what family situation. Lonely fathers, for example, have been found to be at higher risk of CVD and mortality than cohabiting fathers [27]. Neither can marital strain be accounted for, a factor that might influence the number of children, and also independently relate to a higher risk of MACEs, CVD, and premature mortality [37]. Nevertheless, these types of misclassifications related to fertility, family dynamics and other unaccountable factors would rather tend to reduce the difference between the childless men and fathers and thus cannot explain the statistically significant differences reported by us.

Furthermore, in a prospective study, detecting a risk factor at the baseline examination might have given the opportunity to some of the men to adjust their lifestyle or receive treatment and thus possibly influence their risk of developing MACE and diabetes. Our study does not allow any conclusion as to whether the biological background to childlessness is a causative risk factor for CVD and mortality, or whether childlessness is a consequence of a disease which was yet subclinical at baseline.

## Conclusion

This population-based study indicates that male childlessness represents an additional and independent risk factor of all cause and CVD-related mortality, and a risk condition of MACE when more conventional risk factors mediate the risk. This link may be related to psychosocial factors, but it also seems plausible to consider a common underlying biological factor predisposing to infertility and higher mortality risk for example genetics or early life programming. Childless middle-aged men should be considered as target population for prevention, although further research is needed to identify the underlying mechanisms in order to establish cause-related prophylactic measures.

## Author Contributions

**Conceptualization:** Angel Elenkov, Aleksander Giwercman, Sandra Søgaard Tøttenborg, Katia Keglberg Haervig, Peter M. Nilsson.

**Data curation:** Angel Elenkov, Peter M. Nilsson.

**Formal analysis:** Angel Elenkov.

**Funding acquisition:** Aleksander Giwercman, Peter M. Nilsson.

**Methodology:** Angel Elenkov, Aleksander Giwercman, Sandra Søgaard Tøttenborg, Jens Peter Ellekilde Bonde, Clara Helene Glazer, Ane Berger Bungum, Peter M. Nilsson.

**Project administration:** Peter M. Nilsson.

**Supervision:** Aleksander Giwercman.

**Writing – original draft:** Angel Elenkov.

**Writing – review & editing:** Angel Elenkov, Aleksander Giwercman, Sandra Søgaard Tøttenborg, Jens Peter Ellekilde Bonde, Clara Helene Glazer, Katia Keglberg Haervig, Ane Berger Bungum, Peter M. Nilsson.

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
