## [Decision Letter · Decision Letter 0]

17 Dec 2019

PONE-D-19-18778

Male Childlessness as Independent Predictor of Risk of Cardiovascular and All-cause Mortality: A Population-based Cohort Study with more than 30 years follow-up

PLOS ONE

Dear Dr. Elenkov,

Thank you for submitting your manuscript to PLOS ONE. After careful consideration, we feel that it has merit but does not fully meet PLOS ONE’s publication criteria as it currently stands. Therefore, we invite you to submit a revised version of the manuscript that addresses the points raised during the review process.

In this population-based cohort, Angel Elenkov et al evaluated the association between male childlessness and the risk of MACE, cardiovascular mortality, all-cause mortality, and incident diabetes. Authors found that, compared with men with children, childless men were featured with higher fasting blood glucose, triglycerides and blood pressure at the baseline. After adjusting for some co-variables, male childlessness was significantly associated with higher risk of cardiovascular death (HR=1.33) and all-cause death (HR=1.23). Whereas, childlessness was non-significantly associated with lower incidence of diabetes (HR=0.97). Overall, the study is interesting and the quality of this manuscript needs to be improved.

Major concerns:

1.There was no information on why these men didn’t have a child, thus, I wonder whether it was accurate to use the male childlessness as a proxy of impaired fertility.

2.Some important co-variables, such as family history of cardiovascular disease and economic level, should enter into the regression model. Fasting blood glucose, triglycerides and blood pressure should be treated as continuous variable.

3.What’s the rationale of conducting sensitivity analysis? My suggestion is adding the interaction of male childlessness and marital status in the model.

4.Table 1 should present the basic characteristics of people included in the final analysis, not that of entire people of the cohort. Please modify the tables style to make them meet the general requirements of academic publication. 

5.Please check the diagnosis criteria of diabetes.

6.The writing needs to be properly simplified.

We would appreciate receiving your revised manuscript by Jan 30 2020 11:59PM. To enhance the reproducibility of your results, we recommend that if applicable you deposit your laboratory protocols in protocols.io, where a protocol can be assigned its own identifier (DOI) such that it can be cited independently in the future. For instructions see: http://journals.plos.org/plosone/s/submission-guidelines#loc-laboratory-protocols

We look forward to receiving your revised manuscript.

Kind regards,

Ying-Mei Feng

Academic Editor

PLOS ONE

---

## [Author Response · Author response to Decision Letter 0]

29 Jan 2020

Comments to the Author: 

In this population-based cohort, Angel Elenkov et al evaluated the association between male childlessness and the risk of MACE, cardiovascular mortality, all-cause mortality, and incident diabetes. Authors found that, compared with men with children, childless men were featured with higher fasting blood glucose, triglycerides and blood pressure at the baseline. After adjusting for some co-variables, male childlessness was significantly associated with higher risk of cardiovascular death (HR=1.33) and all-cause death (HR=1.23). Whereas, childlessness was non-significantly associated with lower incidence of diabetes (HR=0.97). Overall, the study is interesting and the quality of this manuscript needs to be improved.

Major concerns:

1.There was no information on why these men didn’t have a child, thus, I wonder whether it was accurate to use the male childlessness as a proxy of impaired fertility.

Reply: The issue raised by the reviewer is highly relevant and has also been addressed in the Discussion section of the original version of the manuscript. The group of childless men represent a mixture of men with lowered fertility, those with a sub fertile partner and also subjects who decided not to father children. The registry data we use dates back to the first half of the 1970´s. The possibilities for fertility treatment in men at that time were very limited if present. The modern methods of assisted reproduction are developed in the beginning of the 1990´s. Before that men with a very low fertility potential did not have the possibility to use, for example, intracytoplasmic sperm injection to achieve fertilization and pregnancy. Presumably a big majority of men with infertility before the 1990´s have thus remained childless. We agree that childlessness as a proxy for male infertility has some limitations, for example the fact that we could not account for the participants´ paternal intention, nor the partner’s fertility status, marital strain etc. However, a nation-wide statistical data from Sweden have shown that only one in four childless men are voluntarily childless (Statistics Sweden, SCB: Childbearing patterns of different generations, 2011). This information has now been added to the Discussion part of the manuscript. Fathers of adopted children, or those who have sired offspring with the help of assisted reproduction techniques (ART), might have been included in the control group leading to misclassification bias. Nevertheless, these types of misclassifications related to fertility, family dynamics and other unaccountable factors would rather tend to reduce the difference between the childless men and fathers and thus cannot explain the statistically significant differences reported by us. Furthermore, we regard the 30 years of follow-up, which is the longest available among similar studies, as a major strength of our analysis.

2. Some important co-variables, such as family history of cardiovascular disease and economic level, should enter into the regression model. Fasting blood glucose, triglycerides and blood pressure should be treated as continuous variable.

Reply: We have changed our analysis according to reviewer’s suggestions. We included a separate co-variate “called family history for CVD” (p.8, l.158 - 161): “… A positive family history for CVD was defined as reporting having at least one first-degree relative with MI before the age of 60 years as described previously (24). No family history was defined as no reported first‐degree relatives with MI before 60 years, or no answer in the questionnaire…” 

 We have treated fasting blood glucose, triglycerides and blood pressure as continuous variables. Data on economic level of the subjects is not available from our data set. Our results are not significantly changed and are summarized in the revised table (p.15 l.290 - 298.)

 

3.What’s the rationale of conducting sensitivity analysis? My suggestion is adding the interaction of male childlessness and marital status in the model.

Reply: The rational for including this sensitivity analysis refers to the point of criticism raised by the reviewer in item 1. By excluding men who remained unmarried we reduce the proportion of voluntarily childless men and men without opportunity to reproduce in the “exposed” group and to exclude non-marriage as cause of increased cardiovascular mortality risk in childless men. Our data seem to confirm that this risk increase is related to childlessness per se and not to being married or not. The suggestion of including interaction analysis is interesting, in case the hypothesis that the association between childlessness and increased mortality is modulated by marital status. This was not part of our hypothesis, why such interaction analysis was not included in the original version of the manuscript. However, we have performed these calculations: 

There was a statistically significant interaction between childlessness and marital status in the model evaluating the risk for all-cause mortality (p=0.037) in the partially adjusted model (model 1). However, the interaction was no longer statistically significant in the fully adjusted model 2 (p=0.95). Furthermore, the interaction was not statistically significant in the other analysis evaluating MACEs and CVD mortality in any of the models (data not presented).

As mentioned above, we have omitted this analysis because it was not part of our hypothesis. If the Reviewer/Editor wishes it to be a part of the manuscript, we can easily implement it into the final version. However, for reason mentioned above, we suggest that even the sensitivity analysis is kept as it is now.

4. Table 1 should present the basic characteristics of people included in the final analysis, not that of entire people of the cohort. Please modify the tables style to make them meet the general requirements of academic publication. 

Reply: The table has been modified according to the reviewer’s suggestion (p.12). 

5. Please check the diagnosis criteria of diabetes.

Reply: A technical mistake on line 154: “…had fasting blood glucose > 6.5” was corrected to: “…had fasting blood glucose > 5.6...”

6.The writing needs to be properly simplified.

Reply: We have now taken a second look in the text in order to make the writing concise and simple as much as possible. We would be however grateful if the editors/reviewers point to additional parts of the manuscript which may need re-writing.

---

## [Decision Letter · Decision Letter 1]

22 Apr 2020

PONE-D-19-18778R1

Male Childlessness as Independent Predictor of Risk of Cardiovascular and All-cause Mortality: A Population-based Cohort Study with more than 30 years follow-up

PLOS ONE

Dear Dr. Elenkov,

Thank you for submitting your manuscript to PLOS ONE. After careful consideration, we feel that it has merit but does not fully meet PLOS ONE’s publication criteria as it currently stands. Therefore, we invite you to submit a revised version of the manuscript that addresses the points raised during the review process.

1. Please describe the charactersitics of age, familiy history of CVD, and biochemical factors in the Table 1.  Also, please add the univariate analysis to indicate whether there were significant difference between the two groups in the Table 1. 

2. As the last row of Table 2 shows,  the proportion of elevated BP in the two groups was equal, but the OR was more than 1.0. Please check it.

3. I think Table 2 should show the association of childlessness with metabolic features among participants included in the final analysis (45 years or older), rather than among participants of the entire cohort.  Instead of presenting the reference group using a column, adding this information at the footnote would be better. The column of OR (95% CI) was supposed to be listed on the end of the table. 

4. In Table 3, the three columns of HR (95% CI) should be listed on the right side of the two columns of endpoints incidence. Considering the “total” number of "Childless"/ "Fathers" was fixed in most rows, it should not be marked repetedly, which would simplify the table.

5. Both in Table 2 and Table 3, the first column indicating the different range of paritcipants included in the regression, and please keep the words of the first column consistent.

We would appreciate receiving your revised manuscript by Jun 06 2020 11:59PM. To enhance the reproducibility of your results, we recommend that if applicable you deposit your laboratory protocols in protocols.io, where a protocol can be assigned its own identifier (DOI) such that it can be cited independently in the future. For instructions see: http://journals.plos.org/plosone/s/submission-guidelines#loc-laboratory-protocols

We look forward to receiving your revised manuscript.

Kind regards,

Ying-Mei Feng

Academic Editor

PLOS ONE

---

## [Author Response · Author response to Decision Letter 1]

12 May 2020

Comments raised during the review process:

1. Please describe the charactersitics of age, familiy history of CVD, and biochemical factors in the Table 1. Also, please add the univariate analysis to indicate whether there were significant difference between the two groups in the Table 1. 

2. As the last row of Table 2 shows, the proportion of elevated BP in the two groups was equal, but the OR was more than 1.0. Please check it.

3. I think Table 2 should show the association of childlessness with metabolic features among participants included in the final analysis (45 years or older), rather than among participants of the entire cohort. Instead of presenting the reference group using a column, adding this information at the footnote would be better. The column of OR (95% CI) was supposed to be listed on the end of the table. 

4. In Table 3, the three columns of HR (95% CI) should be listed on the right side of the two columns of 

endpoints incidence. Considering the “total” number of "Childless"/ "Fathers" was fixed in most rows, it should not be marked repetedly, which would simplify the table.

5. Both in Table 2 and Table 3, the first column indicating the different range of paritcipants included in the regression, and please keep the words of the first column consistent.

Reply:

We have made all corrections to all tables as proposed. A number of calculations were also performed with new data included to the tables. The main results and conclusions from our study are not changed.

---

## [Decision Letter · Decision Letter 2]

28 Jul 2020

Male Childlessness as Independent Predictor of Risk of Cardiovascular and All-cause Mortality: A Population-based Cohort Study with more than 30 years follow-up

PONE-D-19-18778R2

Dear Dr. Elenkov,

We’re pleased to inform you that your manuscript has been judged scientifically suitable for publication and will be formally accepted for publication once it meets all outstanding technical requirements.

Kind regards,

Ying-Mei Feng

Academic Editor

PLOS ONE

---

## [Editor Report · Acceptance letter]

19 Aug 2020

PONE-D-19-18778R2 

Male Childlessness as Independent Predictor of Risk of Cardiovascular and All-cause Mortality: A Population-based Cohort Study with more than 30 years follow-up 

Dear Dr. Elenkov:

I'm pleased to inform you that your manuscript has been deemed suitable for publication in PLOS ONE. Congratulations! Your manuscript is now with our production department. 

Kind regards, 

on behalf of

Dr Ying-Mei Feng 

Academic Editor

PLOS ONE